# Association between Wrist Circumference and Risk of Any Fracture in Adults: Findings from 15 Years of Follow-Up in the Tehran Lipid and Glucose Study

**DOI:** 10.3390/jcm11237048

**Published:** 2022-11-29

**Authors:** Seyed Saeed Tamehri Zadeh, Seyyed Saeed Moazzeni, Samaneh Asgari, Mohammadhassan Mirbolouk, Fereidoun Azizi, Farzad Hadaegh

**Affiliations:** 1Prevention of Metabolic Disorders Research Center, Research Institute for Endocrine Sciences, Shahid Beheshti University of Medical Sciences, No. 24, Parvaneh Street, Velenjak, Tehran P.O. Box 19395-4763, Iran; 2Department of Medicine, Yale New Haven Hospital, New Haven, CT 06510, USA; 3Endocrine Research Center, Research Institute for Endocrine Sciences, Shahid Beheshti University of Medical Sciences, Tehran P.O. Box 19395-4763, Iran

**Keywords:** wrist circumference, incident fracture, osteoporosis, major osteoporotic fracture

## Abstract

We evaluated whether wrist circumference (WrC), as a novel anthropometric measure, is associated with incidences of any fractures. The study population included 8288 adults (45.3% men) aged ≥30 years, who were followed for incidences of any fractures from 31 January 1999 to 16 March 2016. We used Cox proportional hazard models adjusted for well-known risk factors to evaluate the association of WrC, both as continuous and categorical variables (bottom tertile as reference), with incidences of any fractures and major osteoporotic fractures (MOF). Over 15 years of follow-ups, 348 fractures occurred (men = 162). For a 1 cm increase in WrC, hazard ratios (HRs) were 1.18 (95% CI: 1.03–1.35) for incident any fractures and 1.22 (1.01–1.49) for incident MOF. In addition to WrC, age, female sex, lower BMI, higher WC, current smoking, and usage of steroidal medications were significantly associated with the incidences of any fractures. Moreover, participants in the middle and top tertiles of WrC had a higher risk of incidence for any fractures [HR = 1.62 (1.19–2.20) and 1.70 (1.14–2.55), respectively, *p*-value for trend = 0.012]. We presented WrC as a strong and independent risk factor for incidences of any fractures that might be considered in the risk prediction of bone fracture in Iranian adults.

## 1. Introduction

During the recent decades, a significant increase in the prevalence of osteoporotic fractures, which primarily stems from improvements in life expectancy, has been observed. Due to the increasing mortality rate and profound loss of quality of life following an osteoporotic fracture, this is an important public health concern [1,2,3]. The lifetime risk of hip, forearm, and vertebral fracture is about 40%, which is similar to the risk of cardiovascular disease [4]. Iran is not exempt from the osteoporosis epidemic. Recently, we found that the annual age-standardized incidence rates of major osteoporotic fractures (MOF), defined as a composite of the fractures that occurred in the vertebral, wrist, hip, and pelvic sites, were about 202 and 342 per 100,000 person-years for men and women, respectively [5].

Up to now, a number of different population studies with the purpose of identifying the risk factors of osteoporosis have been conducted. Although the implication of several clinical risk factors mainly including, but is not limited to, age, current smoking habit, glucocorticoid usage, familial history of hip fracture, alcohol consumption, and vitamin D deficiency for osteoporosis are established, no risk factor is solely capable of predicting the risk of osteoporosis-related fractures, which can attribute to the multi-factorial nature of the disease [2,4,6]. Despite that, there is a mounting imperative for the early detection of individuals who are prone to osteoporosis by utilizing clinical risk factors in order to set up appropriate policies for predicting fractures [6]. Among different anthropometric indices, the association between general and central obesity with osteoporosis and leading fractures was addressed in several studies [7,8]; however, to our best knowledge, no study has examined the association between wrist circumference (WrC) and incident fracture. A recent meta-analysis conducted among a few cohort studies showed that a higher WrC was associated with a greater risk of cardiometabolic disorders among the adult population [9]. Moreover, the associations of cardiometabolic disorders such as glucose intolerance, hypertension, and abdominal obesity with the incidence of fracture were addressed in some studies, with inconsistent results [10,11]. In the current study, we examined the association of WrC with the incidence of any fractures in a population-based cohort study called the Tehran Lipid and Glucose Study (TLGS), during more than a decade of follow-ups.

## 2. Materials and Methods

### 2.1. Study Design

We utilized the data of TLGS, a community-based prospective study on a representative sample of Tehran’s citizens living in district 13. TLGS was conducted to determine the epidemiologic aspect of non-communicable diseases (NCDs) and their risk factors. It also aimed at advancing healthier lifestyles to prevent NCDs.

Details of the study design, recruitment, follow-up, and set-up is published elsewhere [12]. In brief, participants were enrolled in phases I (1999–2001) and II (2002–2005). Then, they were followed for data collection. Further details for the TLGS have been published elsewhere [12].

### 2.2. Study Population

For the current study, among a total of 9747 individuals aged ≥30 years [8071 individuals from phase I and 1676 new participants from phase II], 488 subjects were excluded due to missing information on covariates, including WrC, body mass index (BMI), waist circumference (WC), fasting plasma glucose (FPG), systolic blood pressure (SBP), diastolic blood pressure (DBP), physical activity level, smoking status, marital status, and education level. Then, 768 subjects with no follow-up measurements were also excluded. Finally, 8288 eligible participants [6815 from phase I and 1473 from phase II] remained for our analysis (85% response rate).

### 2.3. Clinical and Laboratory Measurements

Using standard questionnaires, a trained nurse gathered data on demographic characteristics, past medical history, drug history, smoking status, physical activity level, and education level.

While participants were asked to hold the anterior surface of their right wrist up, their WrCs were measured with a tape meter; the tape measure was placed distal to the prominences of the ulnar and radial bones without any pressure over the tape. Weight was measured with shoes removed, with participants wearing light clothing to the nearest 100 g. While shoulders were in normal alignment, the heights of the subjects were measured in the standing position. BMI was calculated as weight divided by the square of the height (kg/m^2^). After a 15-min rest in a sitting position, two measurements of SBP and DBP on the right arm were taken. The mean of these two measurements was considered the subject’s blood pressure (BP). Further details of anthropometric and blood pressure measurements have been published previously [12]. After 12–14 h of overnight fasting, between 07:00 AM and 09:00 AM, blood samples were collected and analyzed on the same day. More details on laboratory data, including FPG, triglycerides, high-density lipoprotein cholesterol (HDL-C), total cholesterol (TC), and creatinine, have been expounded previously [12].

### 2.4. Definition of Terms

Participants were divided into three tertiles according to WrC; bottom tertile: <16.4 cm, middle tertile: 16.4–17.6 cm, top tertile: ≥17.6 cm. Diabetes was defined as having an FPG ≥ 7 mmol/L or taking any glucose-lowering medications. Hypertension was defined as either having an SBP ≥ 140 mmHg, DBP ≥ 90 mmHg, or using anti-hypertensive medications. Education levels were classified as having <6 years, 6–12 years, and >12 years of formal education. Smoking status was categorized into current smokers, former smokers, and never smokers. According to the TLGS protocol, the Lipid Research Clinic (LRC) questionnaire was used for those enrolled in phase I, in which low physical activity was defined as having physical activity less than three days per week. Furthermore, by the Modifiable Activity Questionnaire (MAQ), for those subjects who were enrolled in phase II, having less than 600 min per week was defined as the low physical activity group [12,13]. A positive history of cardiovascular disease (CVD) was considered as having a history of coronary heart disease/stroke. In the current study, menopausal status was defined according to the definition of the World Health Organization (WHO); namely, no spontaneous menstrual periods for at least a year in the absence of any pathological or physiological causes [14]. Moreover, for women with missing information on menstrual bleeding status, being older than 50 years was considered postmenopausal [15].

### 2.5. Outcome Assessment

The details of the data collection of the outcomes have been published before [12]. Briefly, each participant was followed-up under continuous surveillance for any medical complications leading to hospitalization. Based on the TLGS design and protocol, a trained nurse called all participants yearly and recorded any medical events during the last year. A trained general practitioner followed-up any reported event with a home visit and collected complementary medical data from the hospital. Collected data were evaluated by an outcome committee, which included a principal investigator, an internist, an endocrinologist, a cardiologist, an epidemiologist, and other experts, if required. After adjudication by the outcome committee, each event was assigned to a specific outcome. It is of importance to note that the outcome committee did not know the status of the baseline risk factors. Following the TLGS protocol, any fracture, whether in the upper extremities, lower extremities, or other sites, requiring at least a 24 h admission to the hospital were recorded and adjudicated by the outcome committee. As recently reported [5], we ascertained any fracture diagnoses based on the patients’ assertions and hospital discharge summaries. In the present study, we categorized any fractures into four main subgroups: upper extremity, lower extremity, vertebral, and the other fractures. Upper extremity fractures included upper humerus, wrist, hand, clavicle, elbow, and forearm fractures. Lower extremity fractures included pelvis, hip, femur, patella, tibia, fibula, ankle, or foot fractures. Other fractures included ribs, scalp, fascial, or sternum [16,17]. Moreover, in the current study, the fractures that occurred in the vertebral, wrist, hip, and pelvic sites were defined as MOF, as addressed by other investigators [18,19].

### 2.6. Statistical Analyses

Baseline characteristics of the study population were described as mean (standard deviation: SD) values for continuous variables and as frequencies (%) for categorical variables. A comparison of the baseline characteristics across WrC tertiles was made using the ANOVA test for normally distributed continuous variables, the Chi-squared test for categorical variables, and the Kruskal–Wallis statistic for skewed variables. Moreover, a comparison of the baseline characteristics of those with and without outcome occurrence (incident fracture) was performed using the Student’s *t*-test for normally distributed continuous variables, the Chi-squared test for categorical variables, and the Mann–Whitney U statistic for skewed variables.

The crude incidence rate was calculated by dividing the number of new fracture cases by person-years at risk for each sex and total population.

Cox proportional hazard models were applied to evaluate the association with WrC, both as continuous and categorical variables (using the bottom tertile as reference), with an incident fracture in two models: model 1 included sex and age; model 2 was further adjusted for potential risk factors including BMI, WC, diabetes, hypertension, current smoking, education level, low physical activity, positive history of CVD, and steroidal medications usage. Moreover, considering sex stratification, we added menopausal status to model 2 for women. The hazard ratios (HRs) and 95% confidence intervals (95% CI) were reported for WrC and other risk factors. The event date was defined as the date of the incident any fracture. Patients with the following criteria were excluded from our study: leaving the residential area, death, loss to follow-up, or end of follow-up. For participants who experienced incident any fracture, we defined survival times as the period between the entered date and the event date. Furthermore, for excluded individuals, the survival time was defined as the period between the entered date and the latest follow-up date.

We also calculated the interaction of sex with WrC in multivariable models. Since significant interactions were not observed between sex and WrC (at minimum *p*-value < 0.05 for significance), all analyses were performed in the total population. Despite this, we also presented our results of each sex. Moreover, to exclude the probability of fragility fracture, we also examined the interaction between age groups (≥50 and <50 years old) and WrC for incident fractures in multivariable analysis. Since significant interactions were not observed between age group and WrC, the main analysis was performed in the whole population.

Schoenfeld’s global test of residuals was used to test the proportionality assumption of the multivariable Cox regression. All analyses were conducted using STATA version 14 SE (StataCorp, TX, USA) and a two-tailed *p*-value < 0.05 was considered significant.

## 3. Results

The study sample included 8288 participants (men = 3759) aged ≥30 years [mean age (SD) 47.6 (12.5) years]. Baseline characteristics across the WrC tertiles are presented in Table 1. Men had a higher range of WrC. Generally, among continuous variables, cardiometabolic risk profiles became worse with higher WrCs. Thus, being in the top tertile was associated with older age, higher BMI, higher WC, increased BP, lower HDL-C levels, and higher levels of FPG and triglyceride. The prevalence of current smoking, positive history of CVD, hypertension, and diabetes were higher in the top tertile than in the middle and bottom tertiles.

Baseline characteristics of the participants, stratified by the outcome occurrence (incident fracture) during follow-up, are also reported in Table 2. Compared to those without fractures during follow-up, participants who experienced incident fractures had higher ranges of age, WrC, WC, SBP, and TC; moreover, the prevalence of being widowed/divorced, being low-educated, having a positive history of CVD at baseline, and being hypertensive were higher among those with incident fractures.

During a median follow-up of 15.9 years (interquartile range:11.7–16.5), 162 male and 186 female participants experienced an incident fracture. As illustrated in Figure 1, in both sexes, most of the cases belonged to the fractures of lower extremities, including fractures in the patella, femur, hip, pelvis, and leg. The crude incidence rate [95% CI] of incident fracture was 3.0 [2.7–3.4] per 1000 person-years in the total population. The corresponding values were 3.2 [2.7–3.7] in men and 2.9 [2.5–3.7] in women.

Multivariable HRs [95% CI] of incidence of any fractures are shown in Table 3 by considering WrC as a continuous variable. In model 1, adjusted for age and sex, a 1 cm increase in WrC was associated with an HR of 1.12 [1.01–1.24]. Moreover, after further adjustment in model 2, the HR of a 1 cm increase in WrC reached 1.18 [1.03–1.35]. Importantly, in addition to WrC, older age, being female, lower BMI, higher WC, current smoking, and usage of steroidal medications were significantly associated with incident fractures in model 2.

Multivariable HRs [95% CI] of incidence of fracture are shown in Table 4 after considering WrC as a categorical variable. In comparison with the bottom tertile, participants in the middle and top tertiles of WrC had a higher risk of incidence of fracture; the significant risks reached HRs of 1.62 [1.19–2.20] and 1.70 [1.14–2.55] in the fully adjusted model 2, respectively. Moreover, the *p*-values for the fracture risk trend across tertiles were significant in both models (0.024 in model 1 and 0.012 in model 2). Similar to Table 3, older age, being female, lower BMI, higher WC, current smoking, and usage of steroidal medications were found to be significant risk factors in model 2, shown in Table 4.

Multivariable HRs [95% CI] of incident MOF are presented in Table 5. In the fully adjusted model 2, a 1 cm increase in WrC was associated with incident MOFs by an HR of 1.22 [1.01–1.49]. Furthermore, compared to the bottom tertile, having a WrC of 16.4–17.6 cm and ≥17.6 cm (middle and top tertiles) had significant HRs of 1.77 [1.16–2.71] and 1.78 [1.01–3.17], respectively.

To show the robustness of our findings, we performed two sensitivity analyses. As shown in Figure 2 (Panel A), there was no significant interaction between WrC and sex for incident fracture in multivariable analysis (*p*-value for interaction = 0.53); however, these associations were stronger among women. (HRs for middle and top tertiles of WrC were 1.51 [1.00–2.29] and 1.91 [1.21–3.02], respectively.) We also did not find any significant interaction between WrC and age groups for the incidence of any fracture (*p*-value for interaction = 0.69). However, the associations were stronger among the population aged ≥50 years. (HRs for middle and top tertiles were 1.68 [1.17–2.40] and 1.55 [0.98–2.48], respectively) (Panel B).

## 4. Discussion

To the best of our knowledge, our study provided novel evidence that WrC, reflecting as a simple anthropometric measure, was significantly associated with incidences of any fractures, even after adjusting for well-known risk factors, including age, BMI, waist circumference, diabetes, hypertension, current smoking habit, education levels, physical activity level, cardiovascular disease, and history steroid medications. The significant risk of WrC was also found for MOF.

Data on WrC and incidence of fracture remain scarce. In 2004, Brent et al. evaluated the clinical risk factors of hip fracture in elderly white women. They did not find any significant association between WrC and incidence of hip fracture [20]. Biino et al., in a cross-sectional study, also found that among women, gastrointestinal diseases, using steroid medications, and larger WrCs were associated with osteoporosis [21]. However, neither found an association between WrC and fracture.

It has been suggested that WrC is a strong anthropometric index to assess the bone metabolism [22]. Very few studies have proposed that WrC has other potential applicability in clinical practices, and it can be taken into account as a biomarker of insulin resistance, diabetes and prediabetes, cardiovascular risk, hypertension, and obesity [9]. Recently, Luordi et al. found out there was an inverse association between the ratio of adiponectin-leptin and WrC in 280 children with obesity. Adiponectin prompts anti-inflammatory responses in human [23], and in contrast, leptin has been known as an inflammatory cytokine [24]. Leptin has a complex action on bone; it has shown that it can affect bone negatively [25] and positively [26], and its action depends on some factors. In animal studies, the negative effects of increasing the leptin level on bone metabolism have been observed [27,28]. Adiponectin, which secretes from adipose tissues through several mechanisms such as inhibiting osteoclastogenesis, increase in bone mass, and reduction in bone resorption, confers obvious benefits on bone metabolism [29]. Taken together, since individuals with higher WrC levels have potentially lower ratios of adiponectin-leptin, with the lower protective role of adiponectin and greater destructive role of leptin on bone metabolism, it can be speculated that individuals with high levels of WrC are more susceptible to osteoporosis, and as a consequence, osteoporosis-related fractures.

As another underlying pathophysiological mechanism, it was also shown that higher WrC was also associated with a higher risk of metabolic syndrome in the study conducted by Lourdi et al. [30]. Moreover, the association between metabolic syndrome, which affects more than 30% of the Iranian population [31], with the incidence of fracture was also found in some but not all studies [32,33,34]. Individuals with high insulin resistance are more likely to have inflammation in comparison to individuals with low insulin resistance [35]. Mclaughlin et al. expressed that C-reactive protein levels are notably prone to be elevated in subjects with insulin resistance [35], and given its role for bone fracture incidents [36], increasing WrC, a clinical predictor of insulin resistance, through inducing inflammatory status, can cause a rise in rates of bone fracture. Hence, although we adjusted our data analysis for a wide set of covariates, especially both general and central adiposity, FPG, SBP, DBP, and smoking, it might be concluded that a higher risk of WrC for bone fracture might be attributable to insulin resistance or inflammatory markers such CRP that we did not measure in our study.

Abdominal obesity, as measured by WC, was independently associated with incident fracture in our study, consistent with a meta-analysis in this field [7]. However, there is no consistent protocol for the measurement of WC, resulting in a disparity in measuring published studies methods [37]. By contrast, WrC can be measured easily by a uniform protocol with high reproducibility. Thus, WrC could be used as a marker for the risk of fracture.

As a strength, this is the first cohort study demonstrating the positive association between WrC, a simply measured quantity, and incidence of bone fracture among the adult population. Another strength of our study is that we adjusted for several potential essential confounders. As a limitation, we had no precise data to differentiate between traumatic versus osteoporotic fractures of our patients. To resolve the issue of traumatic fracture, we conducted separate analyses according to age (>50 and ≤50 years), assuming the low prevalence of osteoporotic fragility fractures above the age of 50 years; the approach was applied in another study [38]. Accordingly, we did not find effect modifications of age in the association between WrC and incidence of any fracture. Moreover, as reported by Charles et al., 70% of inpatient fractures are potentially osteoporotic [39]. Furthermore, we included the fractures that should be regarded as predominantly osteoporotic. Additionally, the average prevalence of osteopenia and osteoporosis based on dual-energy X-ray absorptiometry (DXA) scan were 35% and 17%, respectively, in Iran in 2012 [40]. Second, this study was performed in Tehran city and therefore, we cannot generalize our findings to other populations. In fact, the current study is a preliminary report, and extra studies are needed in different places of the world, since background ethnicity has an important role in the risk of fracture [41]. Third, we did not have access to the data of vitamin D, parathyroid hormone (PTH), and alcohol consumption. A meta-analysis among 18,531 Iranian individuals demonstrated that the pooled prevalence of vitamin D deficiency (defined as serum 25(OH) D below 20 ng/mL) is 61.97% (52.53%, 71.40%) [42]. The lack of variation of this important risk factor among the Iranian population might not affect the results. Moreover, since alcohol consumption is forbidden in Iran, the data regarding alcohol consumption is not precise.

For the first time, in a prospective cohort study with over 15 years of follow-ups, we found that WrC was independently associated with the incidence of any fracture among the Iranian population. This simple anthropometric measure might be considered a risk prediction of bone fracture among Iranian adults.

## Figures and Tables

**Figure 1 jcm-11-07048-f001:**
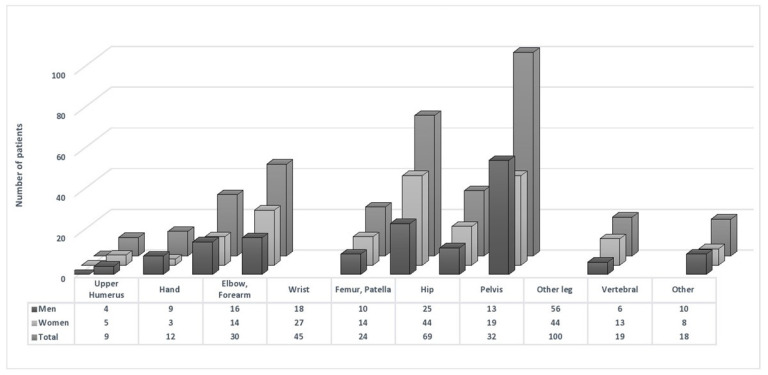
Number of patients across different types of fracture.

**Figure 2 jcm-11-07048-f002:**
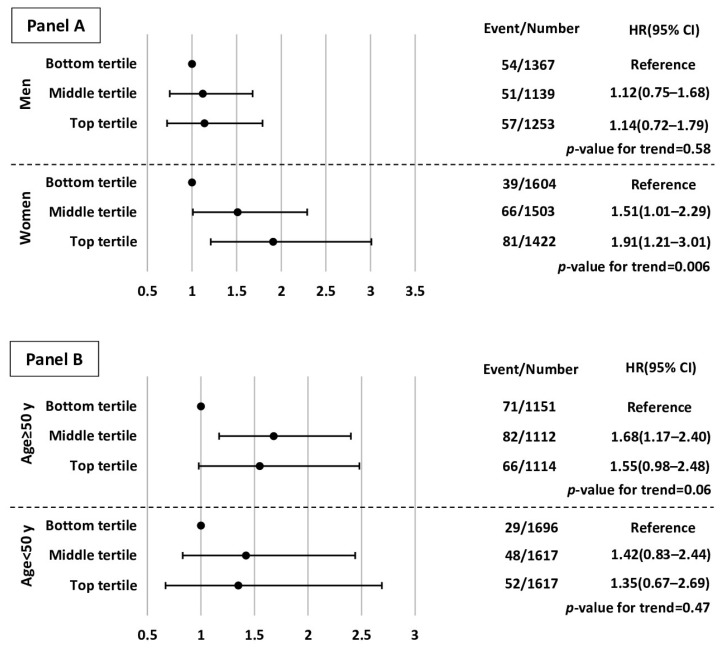
Multivariate hazard ratios (HRs) and 95% confidence intervals (CI) of incident any fractures for wrist circumference stratified based on sex and age: Tehran Lipid and Glucose Study, 1999–2016. y: years. Multivariate hazard ratios were adjusted for sex only in panel (**B**), age only in panel (**A**), body mass index, waist circumference, diabetes, hypertension, current smoking, education level, low physical activity, positive history of cardiovascular disease, and use of steroidal medications. *p*-value for trend (wrist circumference tertiles as a continuous variable in the model). Wrist circumference tertiles measurements in men: [bottom: ≤17.4 cm; middle:17.5–18.1 cm; top: ≥18.2 cm]; in women: [bottom: ≤15.7 cm; middle:15.8–16.5 cm; top: ≥16.6 cm]; in age <50 y: [bottom: ≤16 cm; middle:16.1–17.3 cm; top: ≥17.4 cm]; in age ≥ 50 y: [bottom: ≤16.6 cm; middle:16.7–17.8 cm; top: ≥17.9 cm].

**Table 1 jcm-11-07048-t001:** Baseline characteristics across wrist circumference tertiles: Tehran Lipids and Glucose Study, 1999–2016.

	Total Population	Bottom Tertile (<16.4 cm)	Middle Tertile (16.4–17.6 cm)	Top Tertile (≥17.6 cm)	*p*-Value for Trend *
Number of participants (men)	8288 (3759)	2773 (210)	2996 (1418)	2519 (2131)	
Continuous variables, Mean ± SD					
Age (year)	47.61 ± 12.54	44.43 ± 11.90	48.26 ± 12.29	50.33 ± 12.77	<0.001
BMI (kg/m^2^)	27.49 ± 4.58	26.14 ± 4.02	27.62 ± 4.74	28.85 ± 4.54	<0.001
Waist circumference (cm)	90.81 ± 11.54	85.01 ± 10.55	91.09 ± 10.74	96.84 ± 10.22	<0.001
SBP(mmHg)	121.73 ± 20.02	117.09 ± 19.19	122.72 ± 20.23	125.68 ± 19.67	<0.01
DBP (mmHg)	78.65 ± 11.04	76.49 ± 10.41	78.98 ± 11.03	80.64 ± 11.30	<0.001
FPG (mmol/L)	5.61 ± 1.95	5.42 ± 1.92	5.62 ± 1.93	5.79 ± 1.99	<0.001
Triglycerides (mmol/L)	1.76 (1.31)	1.48 (1.14)	1.81 (1.31)	1.99 (1.33)	<0.001
HDL-C (mmol/L)	1.07 ± 0.28	1.17 ± 0.30	1.06 ± 0.28	0.98 ± 0.24	<0.001
Total cholesterol (mmol/L)	5.54 ± 1.19	5.50 ± 1.22	5.58 ± 1.21	5.52 ± 1.14	0.643
Categorical variables, *n* (%)					
Marital status					<0.001
- Single	357 (4.3)	177 (6.4)	115 (3.8)	65 (2.6)	
- Married	7312 (88.2)	2327 (83.9)	2624 (87.6)	2361 (93.8)	
- Widow/divorced	618 (7.5)	269 (9.7)	257 (8.6)	92 (3.7)	
Current smoker	1386 (16.7)	203 (7.3)	546 (18.2)	637 (25.3)	<0.001
Education level					<0.001
- <6 years	3428 (41.4)	1023 (36.9)	1343 (44.8)	1062 (42.2)	
- 6–12 years	3847 (46.4)	1428 (51.5)	1289 (43.0)	1130 (44.9)	
- >12 years	1013 (12.2)	322 (11.6)	364 (12.1)	327 (13.0)	
Low physical activity	5747 (69.3)	1863 (67.2)	2124 (70.9)	1760 (69.9)	0.029
History of CVD, yes	490 (5.9)	101 (3.6)	193 (6.4)	196 (7.8)	<0.001
Hypertension, yes	2200 (26.5)	549 (19.8)	841 (28.1)	810 (32.2)	<0.001
Diabetes, yes	856 (10.3)	215 (7.8)	314 (10.5)	327 (13.0)	<0.001
Steroidal medications, yes	100 (1.2)	34 (1.2)	35 (1.2)	31 (1.2)	0.993
Any fracture, yes	348 (4.2)	82 (3.0)	144 (4.8)	122 (4.8)	0.001

BMI: body mass index; SBP: systolic blood pressure; DBP: diastolic blood pressure; FPG: fasting plasma glucose; HDL-C: high-density lipoprotein cholesterol; CVD: cardiovascular disease. Values are shown as the mean ± SD and number (%) for continuous and categorical variables, respectively; due to the skewed distribution, for triglyceride, values are shown as the median (interquartile range). * Test for a linear trend according to linear, logistic, or multinomial regressions as appropriate.

**Table 2 jcm-11-07048-t002:** Baseline characteristics of participants, stratified by outcome occurrence (incident fracture) during follow-up: Tehran Lipids and Glucose Study, 1999–2016.

	With Incident Fracture	Without Incident Fracture	*p*-Value *
Number of participants (men)	348 (162)	7940 (3597)	
Continuous variables, Mean ± SD			
Age (year)	53.55 ± 13.13	47.35 ± 12.45	<0.001
Wrist circumference (cm)	17.16 ± 1.20	16.91 ± 1.27	<0.001
BMI (kg/m^2^)	27.41 ± 4.35	27.50 ± 4.59	0.722
Waist circumference (cm)	92.61 ± 10.76	90.73 ± 11.56	0.002
SBP(mmHg)	127.11 ± 21.13	121.50 ± 19.94	<0.001
DBP (mmHg)	79.72 ± 11.36	78.60 ± 11.02	0.073
FPG (mmol/L)	5.81 ± 2.35	5.60 ± 1.93	0.053
HDL-C (mmol/L)	1.10 ± 0.29	1.07 ± 0.28	0.111
Total cholesterol (mmol/L)	5.67 ± 1.25	5.53 ± 1.19	0.036
Triglycerides (mmol/L)	1.74 (1.30)	1.76 (1.31)	0.555
Categorical variables, *n* (%)			
Marital status			0.029
- Single	10 (2.9)	347 (4.4)	
- Married	301 (86.5)	7011 (88.3)	
- Widow/divorced	37 (10.6)	481 (7.3)	
Current smoker	67 (19.3)	1319 (16.6)	0.196
Education level			<0.001
- <6 years	182 (52.3)	3246 (40.9)	
- 6–12 years	134 (38.5)	3713 (46.8)	
- >12 years	32 (9.2)	981 (12.4)	
Low physical activity	255 (73.3)	5492 (69.2)	0.104
History of CVD, yes	30 (8.6)	460 (5.8)	0.029
Hypertension, yes	120 (34.5)	2080 (26.2)	0.001
Diabetes, yes	45 (12.9)	811 (10.2)	0.103
Steroidal medications, yes	8 (2.3)	92 (1.2)	0.057

BMI: body mass index; SBP: systolic blood pressure; DBP: diastolic blood pressure; FPG: fasting plasma glucose; HDL-C: high-density lipoprotein cholesterol; CVD: cardiovascular disease. Values are shown as the mean ± SD and number (%) for continuous and categorical variables, respectively; due to the skewed distribution, for triglyceride, values are shown as the median (interquartile range). * *p*-value for differences between groups with and without incident fracture.

**Table 3 jcm-11-07048-t003:** Multivariable hazard ratios (HR) and 95% confidence intervals (CI) of incidence of any fracture for wrist circumference as a continuous variable: Tehran Lipid and Glucose Study, 1999–2016.

	Model 1		Model 2	
	HR (95% CI)	*p*-Value	HR (95% CI)	*p*-Value
**Wrist circumference (1 cm increase)**	**1.12 (1.01–1.24)**	**0.040**	**1.18 (1.03–1.35)**	**0.016**
**Age, years**	**1.05 (1.04–1.06)**	**<0.001**	**1.05 (1.04–1.06)**	**<0.001**
**Men (women as reference)**	0.88 (0.68–1.15)	0.349	**0.56 (0.39–0.79)**	**0.001**
**BMI, kg/m^2^**			**0.91 (0.86–0.95)**	**<0.001**
**Waist circumference, cm**			**1.03 (1.02–1.05)**	**<0.001**
**Diabetes, yes**			1.09 (0.79–1.51)	0.589
**Hypertension, yes**			0.98 (0.77–1.26)	0.887
**Current smoking, yes**			**1.66 (1.24–2.23)**	**0.001**
**Education level, years**				
- **<6**			**Reference**	
- **6 to 12**			1.28 (0.98–1.68)	0.074
- **≥12**			1.24 (0.82–1.88)	0.306
**Low physical activity, yes**			1.02 (0.80–1.30)	0.871
**Positive history of CVD, yes**			1.13 (0.77–1.66)	0.543
**Steroidal medications, yes**			**2.10 (1.04–4.24)**	**0.039**

BMI: body mass index; CVD: cardiovascular disease. Model 1 was adjusted for age and sex. Model 2 was adjusted for age, sex, BMI, waist circumference, diabetes, hypertension, current smoking, educational level, low physical activity, positive history of CVD, and use of steroidal medications.

**Table 4 jcm-11-07048-t004:** Multivariable hazard ratios (HR) and 95% confidence intervals (CI) of incidence of any fracture for wrist circumference as a categorical variable: Tehran Lipid and Glucose Study, 1999–2016.

	Model 1		Model 2	
	HR (95% CI)	*p*-Value	HR (95% CI)	*p*-Value
**Wrist circumference tertiles**		**0.024 ***		**0.012 ***
- Bottom tertile (<16.4 cm)	**Reference**		**Reference**	
- Middle tertile (16.4–17.6 cm)	**1.51 (1.14–2.01)**	**0.005**	**1.62 (1.19–2.20)**	**0.002**
- Top tertile (≥17.6 cm)	**1.51 (1.07–2.12)**	**0.019**	**1.70 (1.14–2.55)**	**0.009**
**Age, years**	**1.05 (1.04–1.06)**	**<0.001**	**1.05 (1.04–1.06)**	**<0.001**
**Men (women as reference)**	0.88 (0.68–1.13)	0.310	**0.57 (0.41–0.79)**	**0.001**
**BMI, kg/m^2^**			**0.91 (0.86–0.95)**	**<0.001**
**Waist circumference, cm**			**1.04 (1.02–1.05)**	**<0.001**
**Diabetes, yes**			1.09 (0.79–1.50)	0.613
**Hypertension, yes**			0.97 (0.76–1.24)	0.815
**Current smoking, yes**			**1.65 (0.23–2.21)**	**0.001**
**Education level, years**				
**<6**			**Reference**	
**6 to 12**			1.28 (0.98–1.68)	0.073
**≥12**			1.24 (0.82–1.87)	0.316
**Low physical activity, yes**			1.02 (0.80–1.29)	0.902
**Positive history of CVD, yes**			1.10 (0.75–1.62)	0.626
**Steroidal medications, yes**			**2.08 (1.03–4.21)**	**0.040**

BMI: body mass index; CVD: cardiovascular disease. Model 1 was adjusted for age and sex. Model 2 was adjusted for age, sex, BMI, waist circumference, diabetes, hypertension, current smoking, educational level, low physical activity, positive history of CVD, and use of steroidal medications. * *p*-value for the trend (wrist circumference tertile as a continuous variable in the model).

**Table 5 jcm-11-07048-t005:** Multivariable hazard ratios (HR) and 95% confidence intervals (CI) of incidence of major osteoporotic fracture: Tehran Lipid and Glucose Study, 1999–2016.

	Model 1		Model 2	
	HR (95% CI)	*p*-Value	HR (95% CI)	*p*-Value
**Wrist circumference as a continuous variable**				
- 1 cm increase in wrist circumference	1.11 (0.96–1.29)	0.16	**1.22 (1.01–1.49)**	**0.039**
**Wrist circumference as a categorical variable**		0.14 *		**0.041 ***
- Bottom tertile (<16.4 cm)	**Reference**		**Reference**	
- Middle tertile (16.4–17.6 cm)	**1.55 (1.04–2.30)**	**0.03**	**1.77 (1.16–2.71)**	**0.008**
- Top tertile (≥17.6 cm)	1.43 (0.88–2.32)	0.15	**1.78 (1.01–3.17)**	**0.047**

Model 1 was adjusted for age and sex. Model 2 was adjusted for age, sex, body mass index, waist circumference, diabetes, hypertension, current smoking, educational level, low physical activity, positive history of cardiovascular diseases, and use of steroidal medications. * *p*-values for trend (wrist circumference tertile as a continuous variable in the model). Major osteoporotic fracture including wrist, hip, pelvic, and vertebral fractures.

## Data Availability

Data supporting the reported results are available from the corresponding authors upon reasonable request.

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
