# Peer review of "Association between Wrist Circumference and Risk of Any Fracture in Adults: Findings from 15 Years of Follow-Up in the Tehran Lipid and Glucose Study"

_jcm, 2022, doi:10.3390/jcm11237048_

Round 1
Reviewer 1 Report
1. This study was an Interesting prospective cohort study since it arised a good question regarding whether WrC is associated with incidence of any fracture among the adult population. 2. The paper does not explain the clear diagnostic criteria of fracture in adults such as traumatic or osteoporotic fractures,bone density test such as DEXA may be necessary. 3. PTH ,25-OH Vit D, alcohol intake should also be taken into consideration.Author Response
- This study was an Interesting prospective cohort study since it arised a good question regarding whether WrC is associated with incidence of any fracture among the adult population.
Thanks for your precious comment.
- The paper does not explain the clear diagnostic criteria of fracture in adults such as traumatic or osteoporotic fractures, bone density test such as DEXA may be necessary.
Thanks for your comment; agreed, as we mentioned in the limitation part, we had not precise data to differentiate between traumatic versus osteoporotic fracture of our patients. To resolve the issue of traumatic fracture, we conducted separate analyses according to age >50 and ≤50 years, assuming the low prevalence of osteoporotic fragility fractures above the age of 50 years; the approach was applied in other study 1. Accordingly, we did not find effect modification of age in the association between WrC and incidence of any-fracture; however, the associations were stronger among population aged ≥ 50 years (HRs for middle and top tertiles were 1.68 [1.17-2.40] and 1.55 [0.98-2.48], respectively). Moreover, as reported by Charles et al, 70% of inpatients fractures are potentially osteoporotic 2. Furthermore, we included the fractures that should be regarded as predominantly are osteoporotic and excluded scapular fractures since they are mainly due to multi-organ injuries. Additionally, the average prevalence of osteopenia and osteoporosis based on dual-energy X-ray absorptiometry (DXA) scan was 35% and 17%, respectively, in Iran in 2012 3(Introduction; line: 41-43).
- PTH ,25-OH Vit D, alcohol intake should also be taken into consideration.
We did not have access to the data of vitamin D, parathyroid hormone (PTH), and alcohol consumption. Noteworthy, a meta-analysis among 18,531 Iranian individuals demonstrated that the pooled prevalence of vitamin D deficiency (defined as serum 25(OH) D below 20 ng/mL) is 61.97% (52.53%, 71.40%) 4. Lack of variation of this important risk factor among Iranian population might not affect the results. Moreover, since alcohol consumption is forbidden in Iran, the data regarding alcohol consumption is not precise (Introduction; line: 51-2, Discussion; line: 359-365).
- Dominic E, Brozek W, Peter RS, et al. Metabolic factors and hip fracture risk in a large Austrian cohort study. Bone reports. 2020;12:100244.
- Court-Brown CM, Caesar B. Epidemiology of adult fractures: a review. Injury. 2006;37(8):691-697.
- Irani AD, Poorolajal J, Khalilian A, Esmailnasab N, Cheraghi Z. Prevalence of osteoporosis in Iran: A meta-analysis. Journal of research in medical sciences: the official journal of Isfahan University of Medical Sciences. 2013;18(9):759.
4. Tabrizi R, Moosazadeh M, Akbari M, et al. High prevalence of vitamin D deficiency among Iranian population: a systematic review and meta-analysis. Iranian journal of medical sciences. 2018;43(2):125

Reviewer 2 Report
In my opinion, the article is interesting, but firstly it is in fact a preliminary report, secondly, it should be viewed as a picture of a selected part of the Iranian population.
At the outset, the authors write that an osteoporotic fracture, this is an important public health concern, which is true, and that the lifetime risk
of hip, forearm, and vertebral fracture is about 40%, which is also true.
A completely different problem, however, is a pelvic fracture or a scapula. They most often occur in the case of multi-organ injuries. I have doubts whether the combination of multi-organ injuries and osteoporotic fractures is really justified.
Despite these and several other reservations, the work seems interesting to me as the beginning of some discussion and a search for additional indicators of fracture risk.
It seems to me that this should be the beginning of similar research in different places of the world.
It seems to me that the authors should distinguish more what not only results from statistics, but also seems obvious, from what requires further analysis.
Author Response
In my opinion, the article is interesting, but firstly it is in fact a preliminary report, secondly, it should be viewed as a picture of a selected part of the Iranian population.
Thanks for your valuable comment, we added the points you mentioned (Discussion; line: 353, 354, 357, Abstract; line: 31-32).
At the outset, the authors write that an osteoporotic fracture, this is an important public health concern, which is true, and that the lifetime risk of hip, forearm, and vertebral fracture is about 40%, which is also true. A completely different problem, however, is a pelvic fracture or a scapula. They most often occur in the case of multi-organ injuries. I have doubts whether the combination of multi-organ injuries and osteoporotic fractures is really justified.
Thanks for you valuable comment. Regarding pelvic fractures, in the past two decades, it has been suggested to consider pelvic fractures as osteoporotic fractures 2,5 . Parkkari et al. found substantial increase in the incidence of osteoporotic pelvic fractures and proposed to consider pelvic fractures as osteoporotic fractures 6. In another study, it was shown that near two third of all pelvic fractures are osteoporotic and near 94% of pelvic fractures in those aged > 60 years are defined as osteoporotic fractures 7. However, regarding scapula fracture, we are totally agreed with you and the data of scapula fractures were removed from the study. However, we did not find any change in our findings following removing scapula fractures.
Despite these and several other reservations, the work seems interesting to me as the beginning of some discussion and a search for additional indicators of fracture risk. It seems to me that this should be the beginning of similar research in different places of the world. It seems to me that the authors should distinguish more what not only results from statistics, but also seems obvious, from what requires further analysis.
Thanks for you precious comment.
References:
- Court-Brown CM, Caesar B. Epidemiology of adult fractures: a review. Injury. 2006;37(8):691-697.
- Breuil V, Roux CH, Carle GF. Pelvic fractures: epidemiology, consequences, and medical management. Current opinion in rheumatology. 2016;28(4):442-447.
- Parkkari J, Kannus P, Niemi S, et al. Secular trends in osteoporotic pelvic fractures in Finland: number and incidence of fractures in 1970–1991 and prediction for the future. Calcified tissue international. 1996;59(2):79-83.
7. Kannus P, Palvanen M, Niemi S, Parkkari J, Järvinen M. Epidemiology of osteoporotic pelvic fractures in elderly people in Finland: sharp increase in 1970–1997 and alarming projections for the new millennium. Osteoporosis international. 2000;11(5):443-448
